# The Use of Cardiac Stereotactic Radiation Therapy (SBRT) to Manage Ventricular Tachycardia: A Case Report, Review of the Literature and Technical Notes

**DOI:** 10.3390/jpm12111783

**Published:** 2022-10-28

**Authors:** Salvatore Cozzi, Nicola Bottoni, Andrea Botti, Valeria Trojani, Emanuele Alì, Sebastiano Finocchi Ghersi, Federica Cremaschi, Federico Iori, Patrizia Ciammella, Mauro Iori, Cinzia Iotti

**Affiliations:** 1Radiation Oncology Unit, Azienda USL-IRCCS di Reggio Emilia, 42123 Reggio Emilia, Italy; 2Department of Cardiology, Arrhythmology Center, Azienda USL-IRCCS di Reggio Emilia, 42123 Reggio Emilia, Italy; 3Medical Physics Unit, Azienda USL-IRCCS di Reggio Emilia, 42123 Reggio Emilia, Italy; 4Radiation Oncolgy Unit, AOU Sant’Andrea, Facoltà di Medicina e Psicologia, Università La Sapienza, 00185 Rome, Italy; 5Engineer Clinical Specialist, Biosense Webster, Pratica di Mare, Pomezia, 00071 Rome, Italy

**Keywords:** stereotactic body radiotherapy, radioablation, arrhythmias, atrial fibrillation, ventricular tachycardia, ventricular, fibrillation

## Abstract

Background: among cardiac arrhythmias, ventricular tachycardia (VT) is one that can lead to cardiac death, although significant progress has been made in its treatment, including the use of implantable cardioverter-defibrillators (ICD) and radiofrequency catheter ablation. Nevertheless, long-term recurrence rates remain in about half of patients and drastically impact the patient’s quality of life. Moreover, recurrent ICD shocks are painful and are associated with higher mortality and worsening of heart failure. Recently, more and more experiences are demonstrating potential efficacy in the use of stereotactic body radiotherapy (SBRT) (also called cardiac radio-ablation) to treat this condition. In this paper, we report our experience in the use of cardiac radio-ablation for the treatment of refractory ventricular tachycardia with a focus on the technique used, along with a review of the literature and technical notes. Case presentation: an 81-year-old male patient with a long history of non-ischemic dilated cardiomyopathy and mechanical mitral prosthesis underwent a biventricular cardioverter defibrillator implant after atrial ventricular node ablation. At the end of 2021, the number of tachycardias increased significantly to about 10 episodes per day. After failure of medical treatment and conventional RT catheter ablation, the patient was treated with SBRT for a total dose of 25 Gy in a single session at the site of the ectopic focus. No acute toxicity was recorded. After SBRT (follow-up 7 months) no other VT episodes were recorded. Conclusion: SBRT appears to be safe and leads to a rapid reduction in arrhythmic storms as treatment for VT without acute toxicity, representing one of the most promising methods for treating VT storms.

## 1. Introduction

Part of the large family of cardiac arrhythmias, ventricular tachycardia (VT) is a condition that can lead to cardiac death if not properly treated. The impact of VT is growing and has progressively increased. Although significant progress has been made in the treatment of ventricular tachycardia, including the use of implantable cardioverter-defibrillators (ICD) and radiofrequency catheter ablations, long-term recurrence rates remain above 50% [1,2,3,4,5,6]. Treatment failure can be caused by a number of factors, including the limited ability of catheter-based techniques to safely penetrate arrhythmogenic substrates deep in the myocardium, the extent of scar, the resistance of scar to ablation effect, and the likelihood of additional cardiac damage leading to new VT inside and outside the current scar. Catheter ablation may also be associated with procedural risks [7]. In addition, recurrent ICD shocks are painful, contribute to impaired quality of life [8,9,10], and are associated with higher mortality and worsening of heart failure [11].

The use of stereotactic body radiotherapy (SBRT) is widely applied in the management of many oncological diseases [12,13,14,15,16] and could represent a promising therapeutic alternative in cases of failed catheter ablation for recurrent VT, though, to date, there remain limited data on the mechanism of action and longterm effects and efficacy of cardiac radiation therapy for ventricular tachycardia management [17]. SBRT utilizes noninvasive high-dose radiation to create transmural lesions that encompass a predefined zone of VT circuitryStereotactic Body Radiation Technique (or cardiac radioablation) employs radiation delivered to a 3D target volume created by various imaging technologies, accurately ablating the target volume. Furthermore, in contrast with catheter ablation, SBRT is noninvasive.

Early clinical data have demonstrated significant reduction in refractory VT burden with minimal acute and subacute complications [18,19,20,21,22,23,24], although the exact mechanisms by which radioablation damages the target myocardium during treatment of refractory VT have not been fully elucidated.

The purpose of this paper is to report our experience in the use of SBRT for the treatment of refractory ventricular tachycardia (stereotactic arrhythmia radioablation [STAR]), focusing on the technique approach along with a review of the current literature, in view of our participation in the important prospective study organized in Europe which goes under the name of STOPSTORM.

## 2. Case Presentation

An 81-year-old male patient with a long history of non-ischemic dilated cardiomyopathy and mechanical mitral prothesis underwent a biventricular cardioverter defibrillator implant (CRT-D) after atrial ventricular node ablation. In 2016 and 2021, two conventional ablation treatments were performed for sustained ventricular tachycardia. The echocardiography showed a severely enlarged left ventricle with a 20% ejection fraction and an akinetic postero-inferior wall characterized by an aneurysmal evolution. The right ventricle presented a mild hypokinesia while no coronary artery where both free from lesions. The mechanical valve worked normally (Mg 5 mmHg) and no pleural effusion was detectable. Slight tricuspid insufficiency with PAPs of 40 mmHg was recorded.

In January 2022, the patient was hospitalized following two repeated syncopal episodes caused by VT episodes, resulting in head trauma, interrupted by device shock after ineffective antitachycardia pacing (ATP).

Figure 1 shows the number of ventricular tachycardia episodes recorded by the CRT-D. In 2021, the number of ventricular tachycardia episodes was approximately 10 per month. At the end of the year, the number of tachycardias increased significantly to about 10 episodes per day. During hospitalization, recurrent VT episodes were recorded, despite the administration of amiodarone, mexiletine and beta-blockers. 

Considering the ineffectiveness of the treatments used, the patient finally underwent SBRT treatment for a total dose of 25 Gy in a single session at the site of the ectopic focus. No acute or late toxicity was recorded. 

Patient follow-up was carried out through continuous remote monitoring of the device and clinical examination after 1 and 3 weeks and monthly thereafter. Following the radiotherapy treatment session, the device reported reasonable improvement with the complete absence of VT episodes. Seven months after the stereotactic procedure, the patient is alive and in good health, the cardiological picture is stable, and no other syncopal episodes or VT have been recorded. Furthermore, no acute or late cardiac or pulmonary toxicity has been reported.

## 3. Target Volume Definition and Imaging Management

Three-dimensional electroanatomical mapping (EAM) of the chambers of interest was carried out using the CARTO^®^ 3 System (Biosense Webster, Diamond Bar, CA, USA) with dedicated multipolar intracardiac catheters. On the 3D map obtained, anatomical landmarks (the aortic bulb, the left outflow tract and the mitral valve) were identified and labelled. To localize the area responsible for sustaining the arrhythmia, an activation map focusing on the wavefront propagation was performed. EAM points in the target zone were marked to nominal tags and all the CARTO maps were exported. Within a week, the patient underwent electrocardiography (ECG)-gated contrast-enhanced CT (4DCT) and simulation CT without contrast agent (SimCT) on a SOMATOM Confidence^®^ 64S CT scanner (Siemens Healthcare GmbH, Erlangen, Germany). 

Next, using the open-source 3D SLICER software (version 4.11 with EA Map Reader package), the EAM with the marked ablation target were projected onto 4DCT images by landmark registration, with careful manual adjustment (Figure 2). A contour of the scar was performed on the 4DCT, exported to the treatment planning system (version 13.7, Varian Medical Systems, Inc., Varian, Palo Alto, CA, USA) and co-registered with SimCT. Through the combination of these sets of data and following interdisciplinary discussions among the specialists involved, the target area was delineated on the SimCT. Organs at risk such as the esophagus, pericardium, walls of the ventricles and atria, heart valves, aorta and pulmonary arteries and vena cava, the coronary arteries, and finally the stomach were contoured. 

## 4. Treatment Planning and Treatment Delivery

From a previously acquired 4DCT, we estimated the motion range taking into consideration both cardiac and respiratory motion, following the movement of the ICD leads used as reference. The detected range was below 6.4 mm in every direction. After localizing the scar region and using the information about the estimated movement, the internal target volume (ITV) was created from the scar, for a total volume of 57.3 cc. The planning target volume (PTV) was created adding an isotropic margin of 5 mm to the ITV to account for delivery uncertainties, following our institution’s standard practice. The PTV volume was 122.5 cc. The treatment plan consisted of 4 coplanar dynamic conformal arcs (angles used: 330–180°) delivered with 6 MeV flattening filter-free (FFF) photon beams. The dose distribution obtained is reported in Figure 3. A dose of 25 Gy in a single fraction was prescribed to the PTV, normalizing so that 95% of the PTV volume was covered by the 100% isodose (PTV D2% 27.4 Gy, D98% 24.5 Gy, D50% 26.4 Gy, and CTV area D2% 27.4 Gy, D98% 25.9 y, D50% 26.7 Gy). Maximum doses for organs at risk (OAR) outside the heart were kept below the established single-fraction dose limits, following the AAPM TG 101 guidelines together with recommendations from the RAVENTA protocol [25,26].

In terms of cardiac substructures, the left anterior ascending artery (A_LCX) overlapped with the PTV and received a D2% of 19.8 Gy, while the whole heart received a D50% of 5.7 Gy. The electronics of the ICD received a maximum dose of 17.5 Gy due to its close proximity to the PTV. Pre-treatment plan quality assurance was conducted according to internal guidelines for stereotactic radiotherapy. The patient was in a supine position with elevated arms. No vacuum bag or abdominal compression could be used due to the poor clinical condition of the patient. The radiotherapy treatment was performed on 2 February 2022, under close cardiologic ECG monitoring and an emergency stand-by team, using a TrueBeam STx Linear Accelerator (Varian Medical Systems, Inc.). The ICD was kept on during the whole procedure without special reprogramming, according to the national guidelines for 6 MeV photon irradiation without the ICD in the main beam line [27].

Cone beam CT image guidance using the ICD leads as reference for image registration was conducted before the first dynamic arc. Patient positioning and setup of the monitoring equipment took approximately 40 min, while the treatment time including image guidance was approximately 15 min. Beam on time was about 5 min. The treatment was delivered in free breathing.

## 5. Clinical Experiences: Outcomes and Safety

A literature search was performed on the main search engines. To date, 21 studies have been published concerning the use of SBRT in the treatment of ventricular tachycardia and a total of 96 patients have been recorded. Table 1 summarizes the main published studies and treatment outcomes. Three prospective phase I/II studies are available, while the rest are retrospective, of which 11 are case reports. The median age was 75 years (range 29–81). Most of the patients were male (89), while only 7 (7.3%) women were recorded. With the exception of 2 studies, the prescription dose was 25 Gy in a single session. The PTV size ranged from 21 to 303 cc. The follow-up ranged from 2 to 28 months, and only 3 studies reported a follow-up greater than 12 months.

Cuculich et al., in 2017 [20] treated 5 patients with doses of 25 Gy in single fraction. During the 3 months before SBRT, the patients presented with a combined 6577 of VT episodes at 6 weeks post-SBRT, with a baseline reduction of 99.9%. Robinson et al. [22] in a phase I/II trial enrolled 19 patients with ischemic and non-ischemic cardiomyopathy. After cardiac radio-ablation they reported that the median number of episodes was reduced from 119 to 3 (*p* < 0.01), and overall survival was 89% at 6 months and 72% at 12 months.

A study by Lloyd et al. [32] in 2019 treated 10 patients, in which VT was triggered by ischemic failure in 40% and non-ischemic cause in 60%. The decrice of ICD shocks was 68% (2.9 vs. 0.9 shocks/month). Side effects included one patient who had to be resuscitated during procedure and two who had moderate pneumonitis. Three patients underwent to cardiac transplant after SBRT. Microscopic analysis of the explant showed edema and vacuolization with moderate fibrosis.

Neuwirth et al. [19] treated ten patients. With a median follow-up of 28 months, VT episodes were reduced by 87.5% (*p* = 0.012). Of the 10 patients, 2 experienced recurrences of VT up to 3 and 6 months after SBRT.

In most of the studies reviewed, the effectiveness of the treatments was immediate after the radiotherapy treatment. Acute effects mainly occurred within 6 weeks and included valve, early coronary, pericardial, intramyocardial, phrenic nerve, esophageal, and lung damage.

In the study by Cuculich et al. [20], there were no toxicities during SBRT. One patient passed away for a fatal stroke three weeks after SBRT. Three patients reported fatigue the day after SBRT. In the study by Neuwirth et al. [19], three patients experienced ventricular storm, four had nausea, and one mitral regurgitation. Three patients passed away; the cause was vascular dementia with Alzheimer’s disease in one patient and heart failure in the other two. In the Robinson et al. [22] ENCORE VT phase I/II trial, two patients reported grade 3 toxicity consisting of heart failure and pericarditis, five asymptomatic pericardial effusion, and two with pneumonitis.

Currently, SBRT is an option under study and development for medically and radiofrequency refractory VT, with an important reduction in episodes in almost of the reports analyzed and moderate toxicity [41].

## 6. Discussion

Antiarrhythmic drugs and invasive catheter ablation procedures are used to treat patients with ventricular arrhythmias in order to reduce the risk of recurrence. Unfortunately, failure of drugs and/or catheter ablation is common, and recurrence remains a critical issue. Conventional VT ablation can be limited by incomplete or difficult target accessibility with subsequent arrhythmia persistence or recurrence after ablation [42].

Over the last 20 years, several preclinical studies on mice and pigs have investigated the role of ionizing radiation in the treatment of VT, demonstrating that radioablation was able to induce some change in electrophysiological properties (voltage/potential amplitude or bidirectional block) [43,44,45,46].

The mechanism of action of cardiac radioablation is still unknown. It is possible that double-strand DNA breaks are induced that may lead to programmed cell death and subsequently fibrosis. It is postulated that inducing dense transmural fibrosis through apoptosis by ionizing radiation is the primary antiarrhythmic mechanism of action due to abolishment of aberrant conduction in surviving fibers, decreasing re-entry [47].

Moreover, Cx43 upregulation post-radioablation was found in preclinical studies. Upregulation of Cx43 after radioablation appeared functional with increased conduction velocity and decreased repolarization heterogeneity, which is consistent with upregulation of functional gap junctions. These data suggest that the antiarrhythmic effect of radioablation might include temporarily enhanced conduction through upregulation of gap junctions in the target area [43,48,49,50]. In consideration of these promising preclinical data, the first treatments were performed on humans in 2015, triggering an ever-growing interest in the use of SBRT for the treatment of this non-cancer pathology. Indeed, Loo et al., described the first case treated with Cyberknife, with a total dose of 25 Gy, reporting a 90% reduction in the VT rate within the first few months and no acute treatment-related toxicity [16]. Since then, several case reports and smaller case series have been published, but relevant questions still remain open with regard to substrate identification and subsequent delineation on planning computed tomography (CT), optimal dose, and long-term outcome, including toxicity. In the last few years, about 100 patients have been treated for VT using radioablation, as reported by two recent literature reviews [51,52].

The conclusions of the aforementioned works are as follows: (1) STAR reduces sustained VT burden during the first 6 months of follow-up in the vast majority of patients with structural heart disease and therapy-refractory sustained ventricular arrhythmias; (2) recurrence of sustained ventricular arrhythmias is common in this population and mortality reached 26% during the median follow-up period of 1 year; (3) STAR appears to be a safe option for therapy-refractory sustained VT; but (4) acute post-radiation electrical storm can be observed in at least 7% of patients. These data were also confirmed in our case; in fact, after the STAR treatment, the patient no longer experienced VT episodes. However, the follow-up is still too short to define late side effects related to the treatment. Although these conclusions are mostly derived from retrospective studies with small case series, two important prospective studies have been published [22,37]. Robinson et al. [22] conducted a phase I/II study enrolling 19 patients with encouraging results: median noninvasive ablation time was 15.3 min (range, 5.4–32.3). Median number of VT episodes was reduced from 119 (range 4–292) to 3 (range 0–31, *p* < 0.001). The frequency of VT episodes was reduced by 75% in 89% of patients. Overall survival was 89% at 6 months and 72% at 12 months. Use of dual antiarrhythmic medications decreased from 59% to 12% (*p* = 0.008). Quality of life improved at 6 months. However, 2/19 patients (10.5%) developed a treatment-related toxicity (one patient developed heart failure exacerbation and one pericarditis). The study by Carbucicchio et al. [36] is also a phase I/II study, reporting similar results. Moreover, the authors concluded by indicating that STAR can be considered an alternative option for the treatment of VT in patients with structural heart disease. It should also be emphasized how radioablation may lead to a reduction in the use of antiarrhythmic drugs with negative inotropic effects and many other side effects (including depressive disorders), which may improve quality of life, also in a palliative setting. Although rare cases of use of the STAR treatment have been published, the number of patients is extremely small, and it is currently not possible to evaluate its effectiveness in this patient setting [53,54].

## 7. Conclusions

This paper reports our first experience of using SBRT in the treatment of VT in a patient refractory to medical therapy and invasive techniques and confirms, such as the literature data, that this treatment is safe and leads to a rapid reduction in arrhythmic storms, without acute toxicity. Due to the short follow-up, it is not possible to define long-term safety. Wider prospective studies are needed to clarify whether STAR can be a real alternative to invasive techniques, and we await the results of the European STOPSTORM study to draw definitive conclusions.

## Figures and Tables

**Figure 1 jpm-12-01783-f001:**
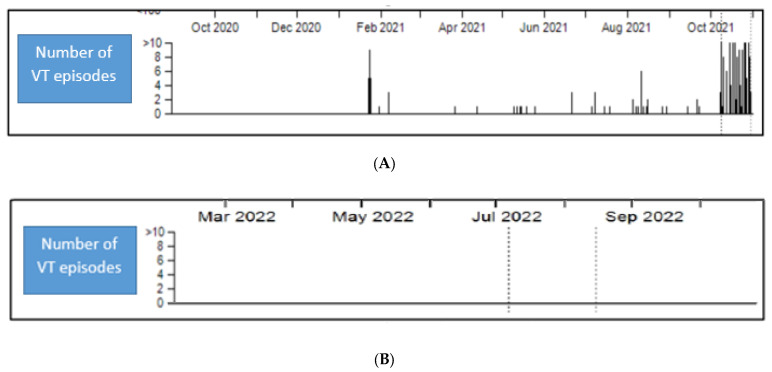
Shows the number of ventricular tachycardia episodes recorded by the biventricular cardioverter defibrillator implant. (**A**) number of ventricular tachycardia episodes before SBRT. (**B**) number of ventricular tachycardia episodes after SBRT.

**Figure 2 jpm-12-01783-f002:**
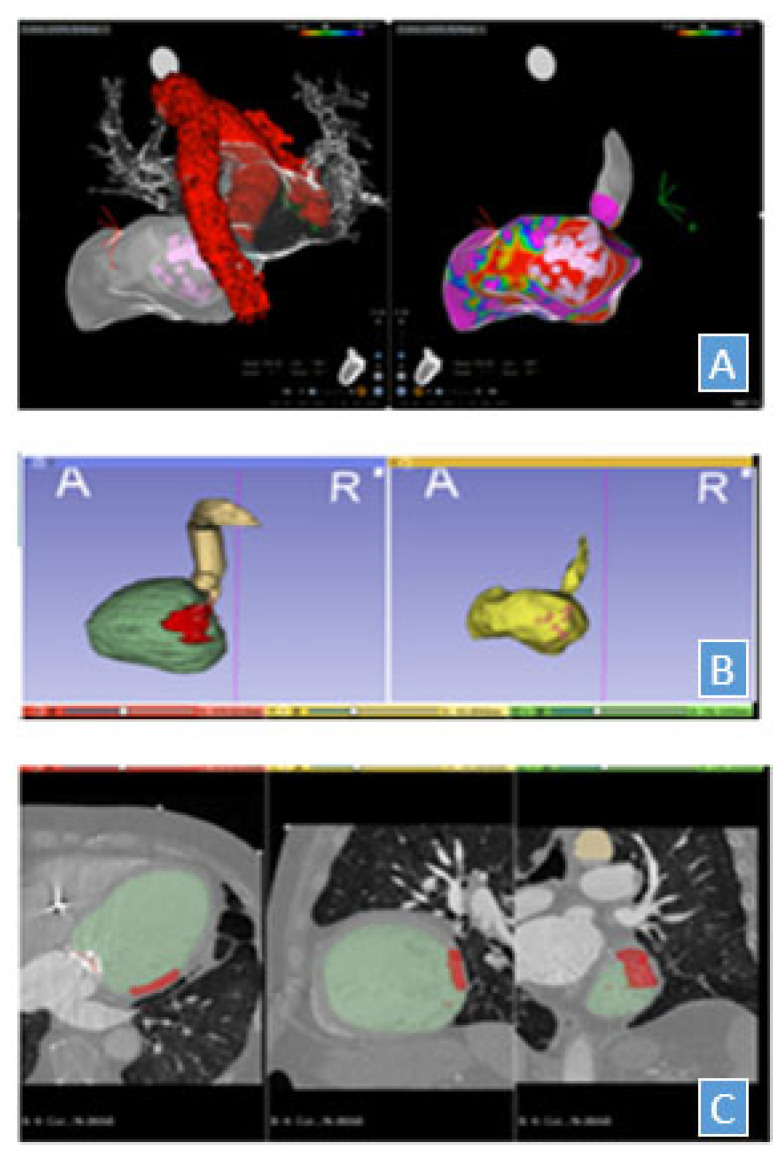
Electroanatomical mapping (EAM) with the marked ablation target were projected onto 4DCT images by landmark registration. (**A**) EAM-the color scale represents a bipolar mapping with extremes of 0.5 and 1.5 mV; (**B**) 3D SLICER software used to project EAM onto 4DTC On the left is presented the rendering of the structures delineated on the CT and registered on the EAM reported on the right; (**C**) Corresponding structures in 4DCT scan.

**Figure 3 jpm-12-01783-f003:**
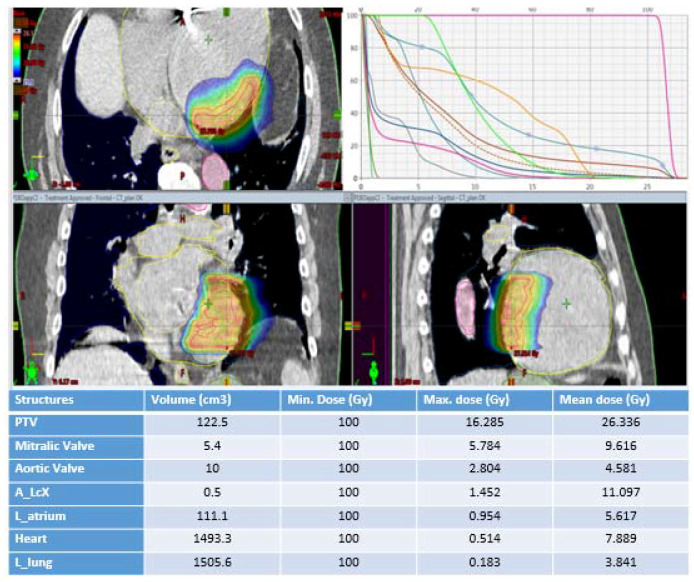
A dose distribution representation of radiotherapy planning and dose volume histogram.

**Table 1 jpm-12-01783-t001:** Summary table of the literature review of published studies.

Study	Type	N of Patients	Sex	Mean Age	Dose	PTV Volume	Follow-Up (m)	Delay for Efficacy	Toxicity
Loo et al., 2015 [18]	R	1	M	71	25	n.r.	9	2 m	Died from COPD
Cuculich et al., 2017 [20]	R	5	4:M 1:F	66	25	49	12	6 w	1 fatal stroke 3 weeks after treatment
Jumeau et al., 2018 [21]	R	1	M	75	25	21	4	immediate	none
Haskova et al., 2018 [23]	R	1	M	34	25	n.r	8	8 m	Not reported
Neuwirth et al., 2019 [19]	R	10	9:M 1:F	66	25	22.2	28	progressive	Three died of non-arrhythmic causes; progression of mitral valve regurgitation at 17 months
Robinson et al., 2019 [22]	phase I/II study	19	17:M 1:F	66	25	98.9	13	within 6 w	Pericarditis; heart failure exacerbation at 2 months
Scholz et al., 2019 [28]	R	1	M	53	30	82.4	2	2 w	none
Zeng et al., 2019 [24]	R	1	M	29	24	71	4	1 m	none
Krug et al., 2019 [29]	R	1	M	78	25	42.2	n.r.	immediate	Cardiac failure after 57 days
Martì-Almon et al., 2020 [30]	R	1	M	64	25	n.r.	4	immediate	none
Magyinger et al., 2020 [31]	R	1	M	71	25	115.1	3	immediate	none
Lloyd et al., 2020 [32]	R	10	7:M 3:F	62	25	81.4	6	within 2 w	2 pts: Mild pneumonitis
Gianni et al., 2020 [33]	P	5	5:M	63	25	143	12	within 6 m	Two died of heart failure
Nauducci et al., 2020 [34]	R	1	M	60	25	303	3	immediate	none
Chin et al., 2021 [35]	R	8	8:M	75	22.2	121.4	8	3 m	none
Carbucicchio et al., 2021 [36]	phase Ib/II study	7	7:M	70	25	183	8	3 m	none
Quian et al., 2021 [37]	R	6	6:M	72	25	319	7.7	6 m	3 patients died of heart failure
Ho et al., 2021 [38]	R	6	6:M	74	25	120.3	6	n.r.	1 pt: pericardial effusion 1 y later
Kautzner et al., 2021 [39]	R	3	M	58	25	61.3	3	immediate	1 pt: myocardial infarction 1 pt: severe bleeding
Molon et al., 2021 [40]	Phase I study	6	5:M 1:F	75	25	n.r.	2–12 m	immediate	2 pt: VT recurrence 1 pt: died of heart failure
Present study	R	1	M	81	25	122.5	6	immediate	none

Abbreviation: N: number; PTV: planning target volume; R: retrospective; M: male; F: female; n.r.: not reported; m: months; w: weeks; COPD: Chronic obstructive pulmonary disease; pts: patients; y: year.

## Data Availability

Not applicable.

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
