# Peer review of "The Use of Cardiac Stereotactic Radiation Therapy (SBRT) to Manage Ventricular Tachycardia: A Case Report, Review of the Literature and Technical Notes"

_jpm, 2022, doi:10.3390/jpm12111783_

Round 1

Reviewer 1 Report

  1. This is a concise and well-written case report with a literature review on a topic of SBRT. Below are few comments that could improve this manuscript

    1. The mechanism of SBRT remains not entirely understood see [PMID.: 34535357]. Therefore paragraph encompassing lines 55-62 is misleading and not supported by available clinical and experimental data. There is no evidence that the histological outcome of SBRT is equivalent to radiofrequency- or cryoablation.
    2. If possible, higher resolution images should be provided, not screenshots but exported and properly labeled images should be used.
    3. It would be great if authors could provide clinical data, for example, echocardiographic data. 
    4. The paper would be strengthened if the authors could discuss how SBRT compared to other arrhythmia treatment modalities, affects contractile function in patients with DCM and or HF. 

Author Response

This is a concise and well-written case report with a literature review on a topic of SBRT. Below are few comments that could improve this manuscript

Dear reviewer thank you for your review. below is the point-by-point answer to your comments

    1. The mechanism of SBRT remains not entirely understood see [PMID.: 34535357]. Therefore paragraph encompassing lines 55-62 is misleading and not supported by available clinical and experimental data. There is no evidence that the histological outcome of SBRT is equivalent to radiofrequency- or cryoablation.

You are definitely right. We have edited the text and added the reference you suggested. a thousand thanks

    1. If possible, higher resolution images should be provided, not screenshots but exported and properly labeled images should be used.

Done. Thanks.

    1. It would be great if authors could provide clinical data, for example, echocardiographic data.

Thank you. We added in case presentation paragraph following sentence: The echocardiography showed a severy enlarged left ventricle with a 20% ejection fraction and an akinetic postero-inferior wall characterized by an aneurysmal evolution. The right ventricle presented a mild hypokinesia while no coronary artery where both free from lesions. The mechanical valve worked normally (Mg 5 mmHg) and no pleural effusion was detectable. Slight tricuspid insufficiency with PAPs of 40 mmHg was recorded.

    1. The paper would be strengthened if the authors could discuss how SBRT compared to other arrhythmia treatment modalities, affects contractile function in patients with DCM and or HF. 

Dear author, your suggestion is extremely interesting, however, based on our knowledge, both from clinical and preclinical studies on animals, we have no data on the impact of SBRT on contractile function. Some studies have postulated a role of SBRT for electrical conduction, however, the evidence is extremely scarce, particularly in patients with DCM or HF. So, in consideration of these scarce  evidences, we have decided not to address the subject.

If you can suggest an article that analyzes this point,  we would be happy to add this topic in the text.

Thank you very much

Reviewer 2 Report

I have read the case report with much interest. Despite it seems to be quite long, the content is appropriate and interesting. Consider shortening of the introduction and discussion sections.

Author Response

I have read the case report with much interest. Despite it seems to be quite long, the content is appropriate and interesting. Consider shortening of the introduction and discussion sections.

Dear reviewer thank you for your review.

We have shortened the introduction. As for the discussion, taking into account that it is not merely a case report, but also an accurate review of the literature and a technical analysis of the method, it is extremely difficult to summarize it further. In our opinion, a further reduction of the text would require making extremely complex concepts unclear for those not accustomed to radiotherpy.